# Uterine Artery Embolization for the Treatment of Symptomatic Uterine Fibroids of Different Sizes: A Single Center Experience

**DOI:** 10.3390/jpm13060906

**Published:** 2023-05-28

**Authors:** Alberta Cappelli, Cristina Mosconi, Maria Adriana Cocozza, Nicolò Brandi, Laura Bartalena, Francesco Modestino, Maria Cristina Galaverni, Giulio Vara, Alexandro Paccapelo, Gloria Pizzoli, Gioia Villa, Renato Seracchioli, Matteo Renzulli

**Affiliations:** 1Department of Radiology, IRCCS Azienda Ospedaliero-Universitaria di Bologna, Via Albertoni 15, 40138 Bologna, Italy; alberta.cappelli@aosp.bo.it (A.C.); cristina.mosconi@aosp.bo.it (C.M.); mariaadriana.cocozza@studio.unibo.it (M.A.C.); laura.bartalena@studio.unibo.it (L.B.); francesco.modestino@aosp.bo.it (F.M.); mariacristina.galaverni@aosp.bo.it (M.C.G.); giulio.vara@studio.unibo.it (G.V.); alexandro.paccapelo@aosp.bo.it (A.P.); matteo.renzulli@aosp.bo.it (M.R.); 2Division of Gynaecology and Human Reproduction Physiopathology, IRCCS Azienda Ospedaliero-Universitaria di Bologna, 40138 Bologna, Italy; gloria.pizzoli@aosp.bo.it (G.P.); gioia.villa@aosp.bo.it (G.V.); renato.seracchioli@unibo.it (R.S.); 3Department of Medical and Surgical Sciences (DIMEC), University of Bologna, 40138 Bologna, Italy

**Keywords:** uterine arterial embolization, uterine myoma, hysterectomy, endovascular embolization, endovascular therapy, interventional radiology

## Abstract

The present study aimed to evaluate the clinical and radiological 1-year outcomes of uterine artery embolization (UAE) performed in a selected population of women with symptomatic myomas and who do not wish to conceive. Between January 2004 and January 2018, a total of 62 patients with pre-menopausal status and with no wish to conceive in the future underwent UAE for the treatment of symptomatic fibroids. All the patients underwent magnetic resonance imaging (MRI) and/or transvaginal ultrasonography (TV-US) before and after the procedure at 1-year follow-up. Clinical and radiological parameters were recorded, stratifying the population into 3 groups according to the size of the dominant myoma (group 1: <50 mm; group 2: ≥50 and ≤80 mm; group 3: >80 mm). Mean fibroid diameter was significantly reduced (42.6% ± 21.6%) at 1-year follow-up, with excellent improvements in terms of both symptoms and quality of life. No significant difference was observed regarding baseline dimension and the number of myomas. No major complications were reported (2.5%). The present study confirms the safety and efficacy of UAE in the treatment of symptomatic fibroids in pre-menopausal women with no desire to conceive.

## 1. Introduction

Uterine fibroids, also known as leiomyomas and myomas, are the most common gynecological benign neoplasia of the reproductive age, affecting 60% of women in this period [1,2,3,4]. The incidence of myomas progressively increases with age until 50 years, when they reach the peak, and then decreases with menopause. Fibroids are more common, tend to present at a younger age, and are greater in number and larger in size in women of African ancestry versus White or Asian women [5]. Women may develop both solitary and multiple uterine fibroids, but the prevalence of the latter is higher [6].

Ultrasonography (US) is the first-line imaging modality for uterine myomas, although it often fails to detect lesions smaller than 1 cm in diameter. Transvaginal US (TV-US) has greater sensitivity than transabdominal US in detecting uterine leiomyomas. However, in consideration of the advantages of magnetic resonance imaging (MRI), this imaging method can be used to precisely determine the number and localization of myomas and even perform volumetric calculations; thus, it is now regarded as the best imaging technique for the study of myomas [7].

Uterine fibroids are generally asymptomatic and found accidentally, although they can cause unbearable symptoms, especially when they reach a considerable size [8]. In fact, patients with uterine leiomyoma might present with pain, heavy menstrual bleeding and/or significant intermittent uterine bleeding with consequent iron-deficient anemia, as well as with bulky symptoms due to the mass effect, including constipation, recurrent urination and hydronephrosis. In addition, uterine fibroids can lead to structural distortion of the uterus and/or reproductive system, [9] which is thought to be responsible for about 1–3% of the overall infertility rate. Additionally, uterine fibroids might cause complications during pregnancy, such as preterm delivery, abnormal fetal position, abnormal placentation, placental abruption, postpartum infections and postpartum bleeding [10].

Due to the related symptoms and complications, myomas can cause significant morbidity and negatively affect fertility, thus impairing the quality of life of these patients and representing an important source of psychological stress. Therefore, the presence of myoma-related symptoms has been one of the primary indications for conducting myomectomy and hysterectomy worldwide [11,12]. For example, uterine fibroids are ultimately the responsible diagnosis for about one-third of the 600,000 hysterectomies performed annually in the USA alone [13].

Over the past 20 years, however, due to the increasing age of first pregnancy, alternative minimally invasive or non-invasive techniques have been largely investigated in order to guarantee clinical symptom relief while still effectively protecting fertility and avoiding uterine removal [14]. Among uterus-preserving therapeutic options, laparoscopic myomectomy is one of the most widely performed interventions. Yet, any surgical intervention carries a small but real risk for complications. Moreover, for patients with myomas that are large, numerous, or unfavorably located, even operative myoma enucleation is associated with the risk of bleeding and hysterectomy. This is especially important in women who are at risk for blood loss and those who wish to keep their uterus for personal or fertility-maintaining reasons [15].

It was not until 1995 that Ravina et al. [16] first proposed uterine artery embolization (UAE) as a valid alternative to surgical treatment of uterine fibroids. UAE is a safe and minimally invasive treatment option that can be performed by interventional radiologists under angiographic guidance. The purpose of the UAE procedure is selective catheterization of the uterine artery and embolization of the arterioles supplying the myoma by using appropriately sized ideal embolic agents, thus cutting off the blood supply to the central and peripheral parts of the myoma [17,18,19]. Moreover, it can also be used as a preoperative support method since it enables the enucleation of large myomas with only minimal blood loss by reducing their size before surgery [20].

A review of the related literature shows significant volumetric reduction rates following UAE, varying between 42 and 83%, with a technical success rate of 96.2% [21,22,23,24]. Moreover, it has been reported that the uterine volume continues to shrink over time, as might be expected as the myomas degenerate [25]. According to the available studies, about 95% of patients experience an improvement in quality-of-life score and symptoms at 1-year follow-up, with only 14.4% of them requiring an additional 3-year procedure and only 4.6% a subsequent surgery (2.8% myomectomy and 1.8% hysterectomy) [21,22]. In addition, as testified by the EMMY trial [26], generic health-related quality of life for women who undergo UAE remains stable even after 10 years, without significant differences compared to patients who undergo a hysterectomy.

Thanks to its minimal invasiveness, UAE has been demonstrated to be superior in terms of shorter hospital stays, anesthetic drug administration, and earlier return to normal activities compared to surgery [27]. In addition to the obvious clinical advantage, these results are of the utmost importance also for the cost-effectiveness of uterine fibroids management. In fact, due to its wide prevalence among the general female population, this disease is a huge economic burden on healthcare systems, reaching a cost of over $2 billion per year in the USA alone [26,28]. In particular, a multicentre study carried out in the U.K. demonstrated that UAE is a cheaper treatment compared to surgical interventions, even after accounting for repeat procedures and complications [29]. However, this initial cost-benefit seems to become neutral at 5 years [30]. Anyways, UAE is certainly associated with a significant decrease in the indirect costs for uterine fibroids treatment, as testified by a recent systematic review, in which the estimated annual cost for lost work (including absenteeism and short-term disability) was significantly lower for those women who underwent UAE compared to those who underwent surgery [31].

Despite its safety and encouraging results, the effects of UAE on pregnancy remain understudied, and its use for women wishing to conceive is still controversial [32,33]. In fact, several studies have reported an increased risk of miscarriage, placental abnormalities, and postpartum hemorrhage following UAE [34], presumably due to induced endometrial ischemia, an inappropriate embolization of the ovaries and/or a distortion of the uterine cavity caused by residual fibroids [35,36,37,38]. Therefore, at least for now, myomectomy still represents the leading treatment for women wishing to conceive [39,40,41].

Since UAE appears to be a very promising treatment, several studies have tried to analyze the pre-procedural factors associated with better outcomes. However, the influence of fibroid size on the success of the UAE is still a matter of debate, and literature data are conflicting. Some earlier reports suggested that larger fibroid size is associated with poorer clinical outcomes and a greater risk of complications due to more extensive collateral circulation [42,43,44]. Conversely, other reports described worse results in smaller myomas, possibly because these are mainly sustained by the collaterals surrounding the uterine walls [45,46,47], and also others found no correlation between baseline uterine dimension and percentage reduction in volume after UAE [48,49,50,51,52].

The present study aims to evaluate the clinical and radiological 1-year outcomes of UAE performed in a selected population of women with symptomatic myomas and who do not wish to conceive. In particular, the study population was stratified into three groups according to myoma’s size to investigate the correlation between baseline diameter and UAE results and, thus, potentially identify those patients who might show better outcomes.

## 2. Materials and Methods

The present study was an observational, retrospective, single-center study and was approved by the local institutional review board (IRB). Informed consent was waived by the Institutional Review Board due to the retrospective nature of the study. The study was conducted in compliance with the Declaration of Helsinki for clinical studies.

### 2.1. Patient Population and Study Design

Between January 2004 and January 2018, a total of 360 patients with symptomatic myomas underwent UAE at the Authors’ Tertiary Center. In particular, to be included in the study, all the patients had to be in pre-menopausal status, had complaints associated with myomas (including menorrhagia, dysmenorrhea, bulky symptoms, and pain), and did not wish to conceive in the future (Figure 1).

All patients were visited by a gynecologist expert in myoma disorders, who confirmed the diagnosis and excluded a viable pregnancy or the presence of an active infection and/or malignancy of the reproductive system, which was all considered absolute contraindications to the UAE procedure and thus exclusion criteria from the study. All patients provided informed consent for the UAE procedure. Contrast allergy, coagulopathy, and renal impairment were considered relative contraindications to the angiographic procedure.

Radiological evaluation of the myomas was assessed through TV-US or MRI independently by two radiologists with 3 and 10 years of experience in the field, both before the procedure (Figure 2) and after 1 year. MRI scans were performed using the same 1.5T MRI superconductive scanner (HDX-t Signa; General Electric^®^, Milwaukee, WI, USA). Pelvic MR scans included pre-contrast axial T1-w images, pre-contrast axial, sagittal, and coronal fat-saturated T1-w images, axial and sagittal T2-w images, axial diffusion-weighted images (DWI), and axial, sagittal, and coronal fat-saturated T1-w images after intravenous administration of gadolinium-based contrast agent (0.1–0.2 mmol/kg; Gadovist ^®^, Bayer-Schering Pharma, Berlin, Germany) [7,53].

In particular, the myoma’s longest diameters and its radiological features (such as the presence of an adenomyosis component) were collected; in the case of multiple myomas, in order to evaluate therapeutic shrinkage, the dominant mass with the largest diameters was chosen as the target lesion. Myomas were classified according to the International Federation of Gynaecology and Obstetrics (FIGO) [54] as class 0 (pedunculated intracavitary), class 1 (submucosal <50% intramural), classes 2–5 (>50% intramural), class 6 (subserosal < 50% intramural), class 7 (subserosal pedunculated), and class 8 (other). Finally, patients were stratified into three groups according to the size of the myoma: group 1: <50 mm; group 2: ≥50 and ≤ 80mm; group 3: >80 mm.

Demographical and clinical symptoms experienced before and after the procedure were obtained from patient files and through telephone interviews and/or during the subsequent gynecological follow-up visit.

According to the Society of Interventional Radiology (SIR) [55], the complications of UAE were divided into minor and major complications. By minor complication, it meant there were no consequences for the patient and, apart from an overnight observation, no therapy was needed; examples of minor complications include post-embolization syndrome (with fever, nausea, vomiting, local pain, and leukocytosis) and severe menstrual cramping (due to fibroid passage and vaginal discharge) By major complication, on the contrary, it meant therapy and hospitalization of the patients were needed and/or the development of permanent sequelae; examples of major complications include contrast media allergic reaction, vessel dissection or aneurysm, arterial thrombosis or pulmonary embolism, ischemic necrotic complications, infections and death [56].

Patients with myomas with an adenomyosis component (i.e., judged responsible for the symptoms) were excluded (*n* = 154), as well as those with incomplete imaging records (*n* = 46). Ninety-seven (*n* = 97) patients were lost during the follow-up; thus, they were excluded.

### 2.2. Peri-Procedural Care

Before the procedure, laboratory blood tests were performed for every patient in order to check renal function, coagulation status, and blood count.

The day before the procedure, the patients were admitted to the Gynecology Department and underwent an anaesthesiologic evaluation; in particular, every procedure was performed with mild conscious sedation using fentanyl and midazolam, whereas pain-controlled analgesia was achieved with either fentanyl or morphine.

On the day of the procedure, patients fasted for at least 8 h and underwent bladder catheterization to prevent discomfort during the procedure and in the post-procedural time. In addition, antibiotic prophylaxis was administered intravenously to prevent post-procedural infections (ciprofloxacin 500 mg).

### 2.3. UAE Procedure

UAE procedure was performed under fluoroscopic guidance and according to the standards of practice [57,58]. In particular, after the administration of local analgesia (Lidocaine 1%), a 5 French (Fr) arterial sheath was placed in the left common femoral artery under ultrasonographic guidance. Then, internal iliac arteries were selectively catheterized using Cobra diagnostic catheter (4 or 5 Fr) (Terumo^®^, Tokyo, Japan) and a 0.035” angled guidewire (Terumo^®^, Tokyo, Japan); a subsequent diagnostic angiogram was carried out to evaluate the anatomy of uterine arteries and to confirm that no vascular anomalies are present and other organs are not affected. Subsequently, the uterine artery is selectively catheterized with a microcatheter (2.7 F) and a microwire (0.021”) (Progreat; Terumo^®^, Tokyo, Japan), and distal embolization was achieved using microspheres 500–700 μm or 700–900 μm in diameter (Embosphere^®^; Merit Medical/Biosphere, Roissy, France) or 500–710 μm nonspherical particles (Contour^®^; Boston Scientific, Natick, MA, USA). Once the embolization is completed on one side, the process is repeated on the opposite side in the same fashion. In the presence of a utero-ovarian anastomosis, a micro-coil was positioned in advance to protect the ovary. At the end of the procedure, a final angiogram was obtained to confirm the correct occlusion of both uterine arteries (Figure 3). The specific embolic agent and the number of particles used were at the discretion of the interventional radiologist, depending on the fibroid size, and the endpoint of the embolization procedure was complete or near complete stasis of blood flow in the uterine artery. The mean number of vials used per procedure was 3.8 (range 1–9), in agreement with the reference works [59].

### 2.4. Post-Procedural Care

After the procedure, patients were discharged one day after bed rest. All patients received ciprofloxacin 500 mg peroral twice daily, non-steroid anti-inflammatory drugs (NSAIDs) and/or minor opioids and proton pump inhibitors for 10 days. The presence of any early complications during the hospital stay was collected, as well as those that emerged during the immediate post-hospitalization period.

### 2.5. Statistical Analysis

Data are reported as means, standard deviations, ranges, and frequencies. The analysis of variance F-test (ANOVA F-test) was used; post hoc analysis was carried out with the Least Significant Difference (LSD) test. All the tests were two-tailed, and a *p*-value of less than 0.05 was considered statistically significant; IBM SPSS 25.0 (SPSS Inc., Armonk, NY, USA) was used to carry out all the statistical analyses.

## 3. Results

During the study period, 62 patients with symptomatic myomas underwent UAE and met the inclusion criteria. The demographical and clinical characteristics of the study population are reported in Table 1. Patient age was not a predictor of UAE response (data not shown).

Of the study population, 62.9% (39/62) patients were diagnosed with multiple myomas, and 37.1% (23/62) presented with a solitary myoma. The type of myoma according to the FIGO classification is reported in Table 2. According to the size of the largest mass, 19.4% (12/62) of patients had a myoma <50 mm (group 1), 40.3% (25/62) a myoma ≥50 but ≤80 mm (group 2), and 40.3% (25/62) a myoma >80 mm (group 3). The average diameter for a single myoma was 85 mm (± 29.8 mm), with the largest lesion treated measuring 150 mm; the average diameter of the dominant lesion in patients with multiple myomas was 67.5 mm (± 28.6 mm), with the largest dominant lesion treated measuring 170 mm (Table 1).

### 3.1. Radiological Evaluation after UAE

The mean uterine fibroid diameter was 74 (±30.1) mm before UAE, which was reduced to 42.2 (±30.1) at the 1-year follow-up after embolization ((mean reduction of 31.8 mm (±20.4)). This is equivalent to 42.6% (±21.6%) overall uterine fibroid diameter response (Figure 4). More specifically, solitary myomas presented a 40.4% (±20.6%) decrease in their diameter, whereas the dominant myoma in patients with multiple masses showed a diameter reduction of 43.8% (±16.4%) (*p* = 0.470) (Table 3).

Despite patients with larger myomas showing a greater reduction in the diameter following UAE, the diameter reduction percentage did not significantly differ between the three groups (Table 4). However, as the pre-procedural diameter increases, a greater reduction of the diameter following UAE is observed (Figure 5).

The number of uterine fibroids showed no positive correlation with dimension reduction following UAE (*p* = 0.711). Despite the greater diameter reduction observed in patients with Class 6 (subserosal < 50% intramural) and Class 0 (pedunculated intracavitary) myomas (40.3% and 31%, respectively), these two groups were also those with bigger lesions at baseline and difference in UAE response according to the different type of fibroid was non-significant.

### 3.2. Clinical Evaluation after UAE

At the 1-year follow-up after UAE, the vast majority of the women treated with UAE referred a significant improvement in their quality of life. Only twelve patients (19%) reported the persistence of symptoms after 1 year, especially abnormal bleeding, although they were reduced compared to baseline. Four of these (6.3%) decided to undergo a subsequent hysterectomy to definitively solve the problem; three cases out of four presented a single intramural myoma > 10 cm (Table 5). The remaining patients with persistent symptoms underwent follow-up and did not undergo a subsequent UAE procedure.

The average length of hospital stay was 4.5 days (±1.5 days). Following the procedure, almost all of the patients (96.8%, 61/63) manifested mild pain, which was promptly controlled with NSAIDs or minor opioids, leaving just a slight abdominal discomfort lasting < 7 days. No patients showed major complications related to the procedure, and only 2 (3.2%) of them had hyperpyrexia (T° > 38 °C) during the hospital stay, which was treated with the administration of antipyretics.

## 4. Discussion

The present 1-year follow-up study confirms the safety and efficacy of UAE for the treatment of symptomatic uterine fibroids in patients less likely to desire fertility but wishing to avoid surgery, as well as in those who are poor surgical candidates. In particular, the procedure allowed a 42.6% overall reduction of myoma diameter, with a significant improvement in the quality of life in the vast majority of patients, due to the disappearance of menstrual or inter-menstrual bleeding (81%) and bulky symptoms (100%). Moreover, thanks to a standardized UAE protocol, pain management, and aftercare available and streamlined by a multidisciplinary team, technical failure was low and comparable to other studies [48], with no major complications.

First of all, the present study supports the evidence that age does not affect the success of the procedure, at least among pre-menopausal women. Previous studies reported that increasing age correlated with decreased volume reduction in leiomyomas [42]. However, in a different study, the same group [49] found that age did not influence fibroid volume change. In a more recent study, Koesters et al. [60] reported that younger patient age and incomplete leiomyoma infarction led to higher rates of treatment failure. Therefore, given the differing outcomes reported in the literature, this question is still not settled.

As the fibroid size is still a matter of debate, the present study attempted to stratify the population into three groups according to myoma’s size (group 1: <50 mm; group 2: ≥50 and ≤80 mm; group 3: >80 mm) to deeper analyze the size reduction following UAE and, potentially, select those patients who might show better outcomes. The major finding of this study was that mean fibroid diameter was significantly reduced by 42.6% at 1-year follow-up, with a rate that is diameter-independent, thus showing no significant differences between small and large fibroids.

Some earlier reports of UAE [42,43,44] suggested that larger fibroid size, as well as a bigger overall uterine volume, are associated with poorer clinical outcomes and a greater risk of complications, with an increased need for hysterectomy. For example, in one study [50], the mean leiomyoma volumetric response was significantly higher in the small leiomyoma group compared with the large leiomyoma group, with a 10.2% greater volumetric response. In particular, UAE for massively enlarged uterine fibroid could be less effective because of more extensive collateral circulation. Despite this evidence, other reports found diametrically opposite results, reporting a positive correlation between baseline leiomyoma dimension and UAE response and thus suggesting a slightly better outcome of larger fibroids [45,46,47]. In particular, they claimed that this different response might be correlated to the different vascularization of small myomas, which is mainly sustained by the collaterals surrounding the uterine walls. Finally, similar to the present study, other Authors found no correlation between baseline uterine dimension and percentage reduction in volume after UAE [48,49,50,51,52]. These conflicting results are difficult to explain for several possible reasons. First, considerable variations in the volume of uterine fibroids and size cut-off were used in the studies. Second, different imaging modalities were used for the evaluation of uterine fibroids, including both MRI and US [49,50], as well as different techniques, such as volumes and diameter [51,61]. Third, many studies, [62,63] like the present one, only included the largest dominant uterine fibroids in patients with multiple leiomyomata, which may have introduced selection bias and some patient-to-patient variation in treatment response. Fourth, different embolization techniques were used in these studies [64] since the number of particles needed to successfully embolize a large uterus may be very different from that needed for a small uterus. Finally, there is variation in the prevalence of extrauterine arterial supply to large leiomyomas. In fact, although larger myomas are more vascular and thus potentially more vulnerable to UAE, [65] subserosal and fundal location might be associated with a collateral ovarian blood supply, reducing the effect of embolization [66,67]. In conclusion, although further investigation to assess the impact of the size, quantity, and type of embolic material in uterine volume changes is still needed, size seems to play a relatively minor prognostic role in patient selection for UAE. The current study, in fact, supports the theory that small fibroids do not necessarily show a better response compared to large ones and vice versa.

Despite one previous study [48] suggesting that patients with single lesions experience a more significant dimensional decrease compared to those with multiple leiomyomas, the number of uterine fibroids did not predict uterine dimension response to UAE in the current cohort. These results are similar to those reported in other studies [49,51,68].

Previous studies observed that the location of leiomyomas might affect outcomes after UAE. For example, some studies reported that submucosal leiomyomas are more likely to undergo a significant change in volume [45,49,69]. Similarly, another one found that pedunculated serosal fibroids were less likely to demonstrate complete infarction following UAE [70]. This evidence is most probably attributed to the difference in vascularity between the different layers of the uterus, with the endometrial layer being more vulnerable to vascular alterations due to the lack of anastomoses among the spiral arteries and the more centrally location (i.e., the most distal layer in relation to the origin of the feeding vessel) [45]. In the present study, despite the greater diameter reduction being observed in patients with class 6 (subserosal <50% intramural) and class 0 (pedunculated intracavitary) myomas (40.3% and 31%, respectively), the percentage of reduction did not significantly differ among groups, similar to other studies but contrary to others [71] This discrepancy with literature data could be explained by the difference in inclusion criteria and in the clinical and radiological characteristics of the study population, as well as by the fact that most previous studies simply divided myomas into submucosal, intramural and subserosal, without using the FIGO classification [54].

In addition to sparing the uterus, UAE offers the advantage of avoiding general anesthesia and surgical complications with a consequent shorter recovery time [72,73], as confirmed by the present results, with an average length of hospital stay of 4.5 days. Almost all the patients of the present study experienced mild post-procedure pain, which is a common side effect of UAE for fibroids, most likely due to the induced ischemia, [74] although without any major complications. In the EMMY Trial [26], a larger fibroid volume was associated with an increased risk of complications. Similarly, other studies [43,75,76] have regarded large uterine fibroids as potential risk factors for rare but serious complications, such as infection and ischemic uterine injury requiring emergent hysterectomy. According to their report, in fact, the extensive infarction of large fibroids could increase necrosis-related complications. Therefore, some experts have questioned whether there is an upper limit for uterine size beyond which UAE should not be recommended. However, the present study, as others in the literature, supports the idea that tumor size is not a risk factor in patients undergoing UAE for fibroids [48,62,77,78].

Despite the encouraging outcomes emerging from the present study, awareness of UAE efficacy for treating symptomatic myomas is still lacking among patients. As testified by an online survey commissioned by the SIR in 2017, 44% of women diagnosed with uterine fibroids had never heard of UAE, and furthermore, 11% of them believed that hysterectomy was the only treatment option. In addition, 73% of women who had heard of UAE did not learn about this treatment option from their gynecologists [79]. Similarly, another survey revealed that the majority of patients presenting for UAE consultation at a community-based Interventional Radiology Practice were self-referred [80]. Finally, a study found that, despite interventional radiologists choosing UAE in five out of seven clinical scenarios, gynecologists chose other treatment options [81]. Therefore, studies that support the effectiveness of UAE for the treatment of symptomatic myomas are still needed in order to highlight the role of this minimally invasive procedure in the management of this invalidating disease.

The present study has several limitations. First of all, the study is retrospective and from a single center; thus, more robust results can be obtained with prospective multicentric studies; however, all the UAE procedures were performed according to the standards of practice. In addition, in the case of multiple myomas, only the patient’s dominant lesion was evaluated and included in the current analysis, which could have introduced bias into the overall treatment response assessment. Moreover, patients were followed up only for 1 year; thus, the long-term outcomes of the procedures, as well as the re-intervention rates, need further analysis. Finally, the study did not assess the quality of life through a standardized and validated questionnaire, although the disappearance of symptoms can be interpreted as the reflection of an excellent result.

## 5. Conclusions

The present study confirms that UAE is a safe and effective alternative procedure to surgery for the treatment of symptomatic uterine myomas. The influence of fibroid size on the outcomes of the UAE is still a matter of debate, and available data are conflicting. The present study demonstrated that UAE outcomes do not depend on fibroid size or number, adding further evidence to the current literature. This procedure, in fact, guarantees a mean dimensional reduction of about 40% at 1-year follow-up of the uterine fibroid diameter, with excellent improvements in terms of both symptoms and quality of life. Moreover, compared to surgery, UAE has the advantages of being minimally invasive and sparing the uterus, with no major complications and a shorter hospital stay, thus reducing the economic burden on healthcare systems. On the basis of these findings, continuing to offer this service to all women with symptomatic fibroids who have no desire to conceive is justified, and the decision on whether to perform it should not depend on the fibroid size. Therefore, multidisciplinary collaboration between gynecologists and interventional radiologists should be implemented accordingly.

## Figures and Tables

**Figure 1 jpm-13-00906-f001:**
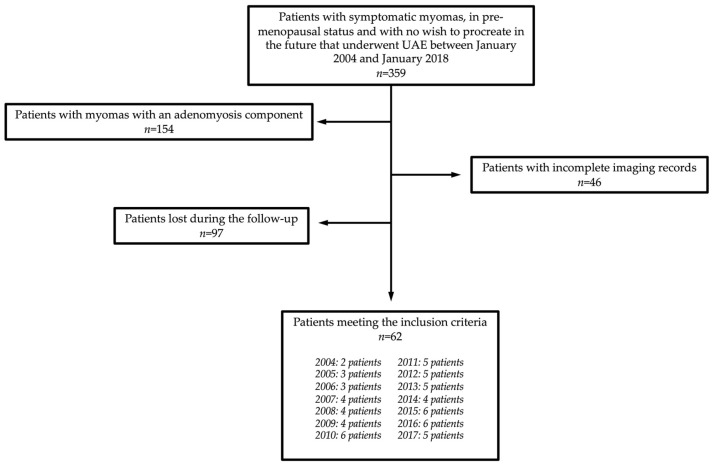
Flow diagram of patient selection in the study.

**Figure 2 jpm-13-00906-f002:**
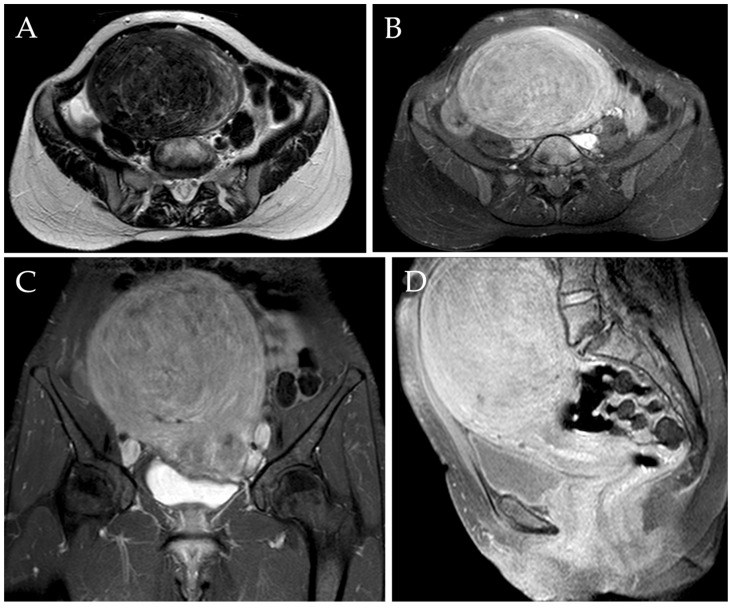
MRI of a 46-year-old woman complaining of metrorrhagia and pelvic pressure. Axial T2-w image (**A**) demonstrated the presence of a voluminous myoma localized in the fundal region, measuring approximately 140 mm. Axial, sagittal, and coronal post-contrast T1-w images with fat saturation demonstrated the typical enhancement of the mass (**B**–**D**).

**Figure 3 jpm-13-00906-f003:**
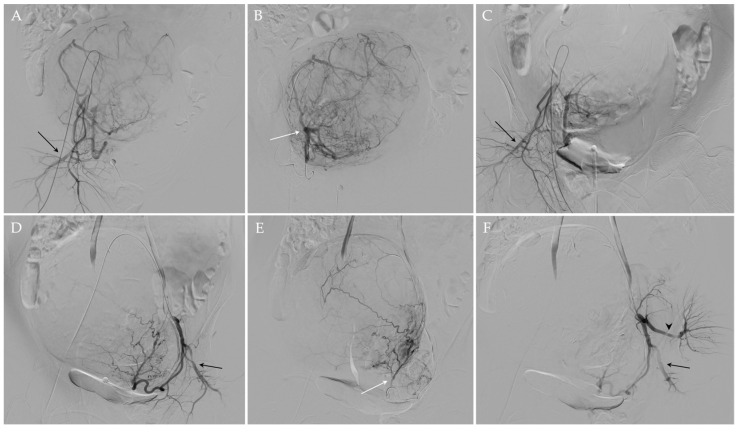
UAE of the same patient. Using a transfemoral approach, the preliminary angiographic study (**A**,**D**) allowed an adequate visualization of the right and left uterine arteries, which appeared hypertrophic, and the right and left pudendal arteries (black arrows in (**A**,**D**), respectively). After the superselective catheterization of the right and left uterine arteries (white arrows (**B**,**E**) respectively), embolization was performed. At the end of the procedure, the occlusion of both uterine arteries was documented and the patency of both pudendal arteries (black arrows in (**C**,**F**)) and of the left gluteal artery (black arrowhead in (**F**)) was appreciated.

**Figure 4 jpm-13-00906-f004:**
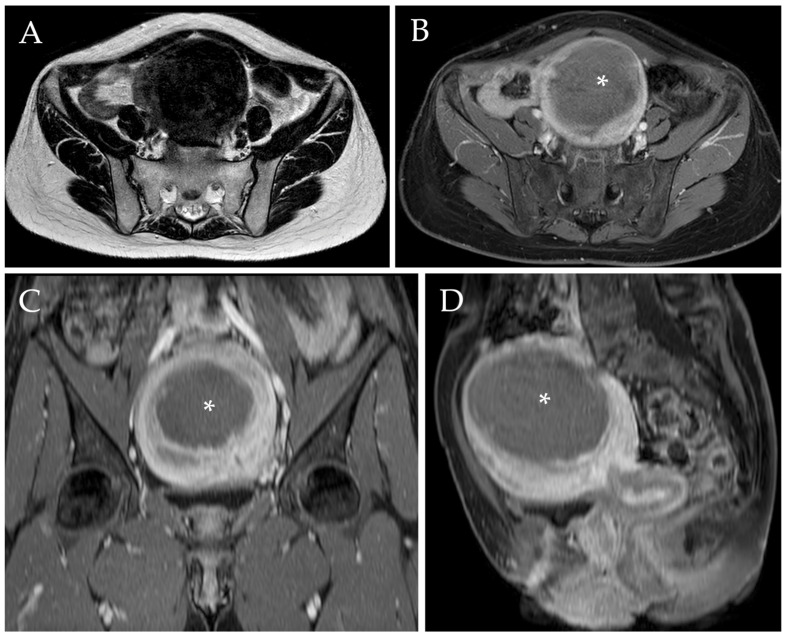
Post-procedural MRI of the same patient. Axial T2-w image (**A**) demonstrated a reduction of the diameter of the voluminous myoma after UAE, which now measures approximately 90 mm (i.e., a diameter reduction of 35.7%). Axial, sagittal, and coronal post-contrast T1-w images with fat saturation (**B**–**D**) confirmed the diameter reduction and revealed the presence of a wide intra-tumoral necrotic area (white asterix).

**Figure 5 jpm-13-00906-f005:**
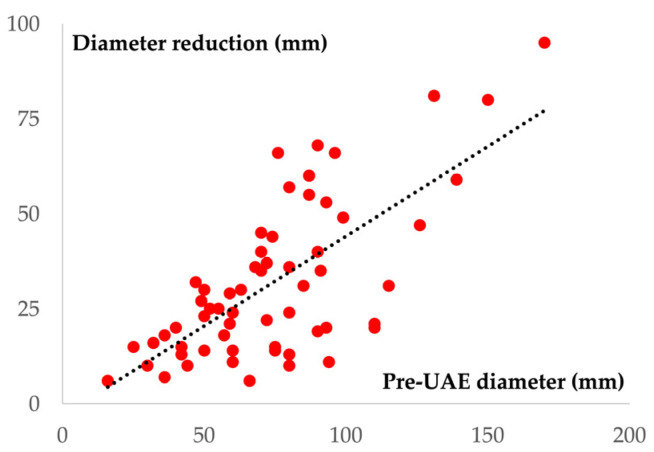
Scatter plot diagram showing a positive linear correlation between the pre-UAE diameter (X-axis) and the reduction of the diameter following UAE (Y-axis) (r^2^ = 0.49; *p* < 0.001).

**Table 1 jpm-13-00906-t001:** Demographical, clinical characteristics, and baseline MRI findings of the study population.

	Total (*n* = 62)
**Age** (mean (±SD))	45 (±3.4)
**Symptoms** (*n* (%))	
Menorrhagia	62 (100%)
Dysmenorrhea	31 (%)
Pain	10 (%)
Bulky symptoms	54 (%)
Obstructive symptoms	4 (%)
**Number of uterine fibroids** (*n* (%))	
Mean number (mean (±SD))	2.3 (±1.3)
Solitary	23 (37.1%)
Multiple	39 (62.9%)
**Diameter of uterine fibroids** (mean (±SD))	
Total	74 (±30.1)
Solitary	85 (±29.8)
Multiple (dominant)	67.5 (±28.6)
**Dimension of uterine fibroids** (*n* (%))	
Group 1 (<50 mm)	12 (19.4%)
Group 2 (≥50 but ≤80 mm)	25 (40.3%)
Group 3 (>80 mm)	25 (40.3%)

SD: standard deviation.

**Table 2 jpm-13-00906-t002:** Type of myoma according to the FIGO classification.

	Class of Myoma According to the FIGO Classification					
	Class 0	Class 1	Class 2–5	Class 6	Class 7	Class 8
**Type of uterine fibroids** (mean (SD))						
Total	1 (1.6%)	2 (3.2%)	43 (68.25%)	16 (25.4%)	0 (0%)	0 (0%)
Solitary	0 (0%)	0 (0%)	21 (91.3%)	2 (8.7%)	0 (0%)	0 (0%)
Multiple	1 (2.6%)	2 (5%)	22 (56.4%)	14 (35.9%)	0 (0%)	0 (0%)

FIGO: International Federation of Gynaecology and Obstetrics; SD: standard deviation.

**Table 3 jpm-13-00906-t003:** Reduction of the size of myoma following UAE at 1-year follow-up according to the type of myoma.

	Pre-UAE Diameter in mm (Mean (±SD))	*p*	Post-UAE Diameter in mm (Mean (±SD))	*p*	Percentage of Reduction (% (±SD%))	*p*
**Total**	74 (±30.1)		42.2 (±30.1)		42.6% (±21.6%)	
**Number of uterine fibroids**		0.026		0.011		n.s.
Solitary	85 (±29.9)		51.1 (±24.5)		40.5% (±20.7%)	
Multiple	67.5 (±28.7)		37 (±18)		43.9% (±16.4%)	
**Type of uterine fibroids**		n.s		n.s		n.s.
Class 0 (pedunculated intracavitary)	115		84		31%	
Class 1 (submucosal <50% intramural)	44.5 (±6.4)		21 (±1.4)		23.5% (±4.9%)	
Class 2–5 (>50% intramural)	70.6 (±24)		41.6 (±20.9)		29.1% (±19.5%)	
Class 6 (subserosal < 50% intramural)	84.3 (±41.2)		42.2 (±21.6)		40.3% (±22.6%)	

UAE: uterine artery embolization; SD: standard deviation; n.s.: non-significant.

**Table 4 jpm-13-00906-t004:** Reduction of the size of myoma following UAE at 1-year follow-up according to the size of the myoma.

	Diameter Reduction in mm (Mean (±SD))	*p*	Percentage of Reduction (% (±SD%))	*p*
**Size of uterine fibroids**				
Group 1 (<50 mm)	15.8 (±7.7)	<0.01	42.7% (±15.1%)	n.s
Group 2 (≥50 but ≤80 mm)	28 (±13.4)		43.3% (±17.8%)	
Group 3 (>80 mm)	43.2 (±23.8)		41.9 (±20.1%)	

UAE: uterine artery embolization; SD: standard deviation; n.s.: non-significant.

**Table 5 jpm-13-00906-t005:** Clinical outcomes following UAE, including length of hospital stay, complications rate, and disappearance of symptoms.

Clinical Outcomes Following UAE	
**Length of hospital stay** (mean days (±SD))	4.5 (±1.5)
**UAE-related complications** (*n* (%))	
Minor	2 (3.2%)
Major	0
**Clinical outcomes following UAE** (*n* (%))	
Disappearance of bleeding	51 (81%)
Regularization of menstrual cycle	48 (76.1%)
Disappearance of bulky symptoms	54 (100%)
Persistence of symptoms	12 (19%)
Subsequent hysterectomy for inadequate symptom control	4 (6.3%)

UAE: uterine artery embolization; SD: standard deviation; n.s.: non-significant.

## Data Availability

All data generated or analyzed during this study are included in this article. Further inquiries can be directed to the corresponding author.

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
