# Peer review of "Uterine Artery Embolization for the Treatment of Symptomatic Uterine Fibroids of Different Sizes: A Single Center Experience"

_jpm, 2023, doi:10.3390/jpm13060906_

Round 1

Reviewer 1 Report

Article: Uterine Artery Embolization for the treatment of symptomatic 2 uterine  fibroids of different sizes: a single center experience. By: Cappelli et al.

In the introduction it mentions very little information on epidemiology,  suggest adding more, this data is relevant for the understanding of the importance of the pathology.

In materials and methods it is mentioned that it was approved by the local institutional review board,  suggest adding the authorization number. 

In figure one, explain why the n was 62 in a period from 204 to 2018,  suggest specifying by year how many patients there were, e.g. 2004: X number of patients, 2005: X number of patients, etc.

Suggest adding in the introduction why it is important to separate the size of the myoma (how relevant it is that it is small, large, etc.) to support why these 3 groups of classification were elaborated in the methodology.

In the conclusion, the knowledge generated is not highlighted, I suggest clearly specifying what was the contribution of this study.

Minor editing of English language required

Author Response

Dear Reviewer,

Please find enclosed the revised version of our manuscript entitled “Uterine Artery Embolization for the treatment of symptomatic uterine fibroids of different sizes: a single center experience” which we request you to consider for possible publication as a Review Article in Journal of Personalized Medicine.

Thank you for the opportunity to revise and improve our paper according to your comments and suggestions.

We have modified the main text in accordance with your insightful and significant suggestions and we have replied point by point to all requested revisions.

We hope that now our manuscript reaches a suitable level for a possible publication in your interesting journal.

The manuscript, approved by all the Authors, has not been published previously and is not under consideration (in whole or in part) for publication elsewhere.

There is no conflict of interest.

We look forward to hearing from you at your earliest convenience.

Sincerely,

Nicolò Brandi and Matteo Renzulli

Department of Radiology, IRCCS Azienda Ospedaliero-Universitaria di Bologna, Via Albertoni 15, Bologna, Italia.

Article: Uterine Artery Embolization for the treatment of symptomatic 2 uterine  fibroids of different sizes: a single center experience. By: Cappelli et al.

In the introduction it mentions very little information on epidemiology, suggest adding more, this data is relevant for the understanding of the importance of the pathology.

RE: Dear Reviewer, thank you for your insightful comments. We have now added a brief sentence to further expand the epidemiological information regarding uterine myomas.

In materials and methods it is mentioned that it was approved by the local institutional review board, suggest adding the authorization number. 

RE: Reviewer thank you very much for the very comment. The UAE procedure is part of the standard clinical care management of our hospital and, therefore, it was not necessary to request the approval of the ethics committee for the execution of this retrospective observational study.

In figure one, explain why the n was 62 in a period from 204 to 2018, suggest specifying by year how many patients there were, e.g. 2004: X number of patients, 2005: X number of patients, etc.

RE: Reviewer thank you very much for the very suitable suggestions. We have modified Figure 1 according to your suggestions.

Suggest adding in the introduction why it is important to separate the size of the myoma (how relevant it is that it is small, large, etc.) to support why these 3 groups of classification were elaborated in the methodology.

RE: Dear Reviewer, thank you very much for the helpful suggestion. We have now added a brief explanation of the current evidence of fibroid volume on UAE outcomes, explaining why the present study aimed to investigate its influence.

In the conclusion, the knowledge generated is not highlighted, I suggest clearly specifying what was the contribution of this study.

RE: Dear Reviewer, thank you very much for the helpful suggestion. We have now specified the contribution of this study to the current literature.

Reviewer 2 Report

Review Report

·     In this article, the authors aimed to evaluate the clinical and radiological 1-year outcomes of uterine artery embolization (UAE)  performed in a selected population of women with symptomatic myomas.

·     Based on a retrospective analysis of 62 women, the authors confirm the safety and efficacy of UAE in the treatment of symptomatic fibroids in pre-menopausal women with no desire to conceive.

·     The paper is interesting. The authors have clearly worked hard to detail their study, but I have some comments:

POINTS OF STRENGTH

1.     Interesting topic.

2.     The results are Ok.

POINTS OF WEAKNESS

1.     Lack of novelty.

2.     Retrospective analysis.

SPECIFIC COMMENTS

1.     Introduction: it is so long…..try to shorten it.

2.      Page 3, line 120: as it is a retrospective design, the patient consent is waived.

3.      Page 3, line 138:  it is better to mention your MRI protocol, including all MRI parameters and the name and dose of contrast used.

4.      What is the number and experience of radiologists who reviewed MR images, and what are their experiences?

5.      Did the image review independently or in consensus?

6.      Post-UAE MR images of the same patient are required to detect the radiological outcomes.

7.      Table 1: the number, diameter and dimension of fibroids; all these are MRI findings and not demographic or clinical data.

8.      Results: avoid repeating the results in the tables. Write the most important result only.

Good

Author Response

Dear Reviewer,

Please find enclosed the revised version of our manuscript entitled “Uterine Artery Embolization for the treatment of symptomatic uterine fibroids of different sizes: a single center experience” which we request you to consider for possible publication as a Review Article in Journal of Personalized Medicine.

Thank you for the opportunity to revise and improve our paper according to your comments and suggestions.

We have modified the main text in accordance with your insightful and significant suggestions and we have replied point by point to all requested revisions.

We hope that now our manuscript reaches a suitable level for a possible publication in your interesting journal.

The manuscript, approved by all the Authors, has not been published previously and is not under consideration (in whole or in part) for publication elsewhere.

There is no conflict of interest.

We look forward to hearing from you at your earliest convenience.

Sincerely,

Nicolò Brandi and Matteo Renzulli

Department of Radiology, IRCCS Azienda Ospedaliero-Universitaria di Bologna, Via Albertoni 15, Bologna, Italia.

In this article, the authors aimed to evaluate the clinical and radiological 1-year outcomes of uterine artery embolization (UAE) performed in a selected population of women with symptomatic myomas.
Based on a retrospective analysis of 62 women, the authors confirm the safety and efficacy of UAE in the treatment of symptomatic fibroids in pre-menopausal women with no desire to conceive.
· The paper is interesting. The authors have clearly worked hard to detail their study, but I have some comments:

POINTS OF STRENGTH

  1. Interesting topic.
  2. The results are Ok.

POINTS OF WEAKNESS

  1. Lack of novelty.
  2. Retrospective analysis.

SPECIFIC COMMENTS

  1. Introduction: it is so long... try to shorten it.

RE: Dear Reviewer, thank you for your appreciation and insightful comments. Although we highly value your suggestion, the other reviewers ask us to add further data regarding the epidemiology of this condition and the rationale behind our study in the introduction section. Therefore, we ask to the Editor if he deems it necessary to reduce the Introduction.

  1. Page 3, line 120: as it is a retrospective design, the patient consent is waived.

RE: Dear Reviewer, thank you for the suggestion. We have now corrected the text regarding the patient’s consent.

  1. Page 3, line 138: it is better to mention your MRI protocol, including all MRI parameters and the name and dose of contrast used.

RE: Dear Reviewer, thank you for the suggestion. We have now specified the MRI protocol and the name and dose of contrast used.

  1. What is the number and experience of radiologists who reviewed MR images, and what are their experiences?
  2. Did the image review independently or in consensus?

RE: Dear Reviewer thank you very much for the very suitable suggestions. Two radiologists, with 3 and 10 years of experience in the field respectively, independently reviewed all the images. We now added this information to the text.

  1. Post-UAE MR images of the same patient are required to detect the radiological outcomes.

RE: Dear Reviewer, thank you very much for the helpful suggestion.  We have now provided pre- and post-procedure MRI of the same patient.

  1. Table 1: the number, diameter and dimension of fibroids; all these are MRI findings and not demographic or clinical data.

RE: Dear Reviewer, thank you very much. You are right. We have now modified the title of Table 1 accordingly.

  1. Results: avoid repeating the results in the tables. Write the most important result only.

RE: Dear Reviewer, thank you very much for your suggestion. We have tried to shorten as far as possible the results section, avoiding repeating the results in the Tables also in the text.

Reviewer 3 Report

Manuscript ID: JPM-2369468
Title
:Uterine Artery Embolization for the treatment of symptomatic uterine fibroids of different sizes: a single center experience 

Date: 2023/5/24

Reviewer's report:
This is an interesting manuscript as it’s a comprehensive study aimed to 
investigate the outcome of  uterine artery embolization (UAE) performed in a selected population of women with symptomatic myomas and who do not wish to conceive. This study was one of the few study  aims to evaluate the clinical and radiological 1-year outcomes of  UAE performed in a selected population of women with symptomatic myomas and who do not wish to conceive. In particular, the study population  investigate the correlation between baseline diameter and UAE results and, thus, potentially identify those patients who might show better outcomes  Final conclusion  confirms the safety and efficacy of UAE in the treatment of symptomatic fibroids in pre-menopausal women with no desire to conceive. I'm sure the result of this study could help in the  decision-making process and guide towards an optimal management for young pre-menopausal patients diagnosed with uterine fibroid.

The MS is well prepared and containing a large amount of data.  Although, there remain some limitation.  Nevertheless, it was still well written. However, a few issue  should be clarify prior publication

1.    What are the contraindication for patient selection aside from no desire to conceive ?

2.    What was the rate of infertility for patients receiving UAE ? 

3.    What was the most effective amount of micron particle use for treating uterine fibroid?

      4.  What was the maximum amount of micron particle for UAE  ? and for larger tumor, how many course of UAE is needed for a complete control of disease ?

Author Response

Dear Reviewer,

Please find enclosed the revised version of our manuscript entitled “Uterine Artery Embolization for the treatment of symptomatic uterine fibroids of different sizes: a single center experience” which we request you to consider for possible publication as a Review Article in Journal of Personalized Medicine. 

Thank you for the opportunity to revise and improve our paper according to your comments and suggestions.

We have modified the main text in accordance with your insightful and significant suggestions and we have replied point by point to all requested revisions.

We hope that now our manuscript reaches a suitable level for a possible publication in your interesting journal.

The manuscript, approved by all the Authors, has not been published previously and is not under consideration (in whole or in part) for publication elsewhere.

There is no conflict of interest.

We look forward to hearing from you at your earliest convenience.

Sincerely,

Nicolò Brandi and Matteo Renzulli

Department of Radiology, IRCCS Azienda Ospedaliero-Universitaria di Bologna, Via Albertoni 15, Bologna, Italia.

This is an interesting manuscript as it’s a comprehensive study aimed to investigate the outcome of  uterine artery embolization (UAE) performed in a selected population of women with symptomatic myomas and who do not wish to conceive. This study was one of the few study  aims to evaluate the clinical and radiological 1-year outcomes of  UAE performed in a selected population of women with symptomatic myomas and who do not wish to conceive. In particular, the study population  investigate the correlation between baseline diameter and UAE results and, thus, potentially identify those patients who might show better outcomes  Final conclusion  confirms the safety and efficacy of UAE in the treatment of symptomatic fibroids in pre-menopausal women with no desire to conceive. I'm sure the result of this study could help in the  decision-making process and guide towards an optimal management for young pre-menopausal patients diagnosed with uterine fibroid.

RE: Dear Reviewer, thank you for your appreciation.

The MS is well prepared and containing a large amount of data.  Although, there remain some limitation.  Nevertheless, it was still well written. However, a few issue should be clarify prior publication

  1. What are the contraindication for patient selection aside from no desire to conceive?

RE: Dear Reviewer, thank you for your insightful comments. We have now specified both absolute and relative contraindications to UAE.

  1. What was the rate of infertility for patients receiving UAE? 

RE: Dear Reviewer, thank you very much for your comment. The main inclusion criterion for the study was no desire to conceive in the future, therefore the rate of infertility among patients receiving UAE was behind the scope of the present study.

  1. What was the most effective amount of micron particle use for treating uterine fibroid?
  2. What was the maximum amount of micron particle for UAE? and for larger tumor, how many course of UAE is needed for a complete control of disease?

RE: Dear Reviewer, thank you very much for your interesting questions. The number of particles used depended on the size of the tumor, as larger fibroids are vascularized by numerous vessels, often very hypertrophic and ramified, which therefore generally required a higher number of particles compared to smaller myomas. Therefore, the optimal number of particles for the procedure is variable from case to case. Anyways, the specific embolic agent used was at the discretion of the interventional radiologist, and the end point of the embolization procedure was complete or near complete stasis of blood flow in the uterine artery. In our study, the mean number of vials used per procedure was 3.8 (range 1–9), in agreement with the reference works [CITA]. In the present study, each patient underwent only one UAE procedure. Only twelve patients (19%) reported the persistence of symptoms after 1 year, especially the persistence of abnormal bleeding. Four of these (6.3%) decided to undergo a subsequent hysterectomy to definitively solve the problem whereas the others are still in follow up. In fact, the latter, despite the persistence, experienced a reduction of symptoms. Therefore, since the main indication for the procedure is the presence of unbearable symptoms rather than the size of the mass, in agreement with the multidisciplinary team, they decided not to undergo a subsequent procedure. These considerations have been specified in the main text.